# Statistical Properties of a Twisted Gaussian Schell-Model Beam Carrying the Cross Phase in a Turbulent Atmosphere

Wenshuo Hou [1,2], Leixin Liu [1,2], Xianlong Liu [1,2], Yangjian Cai [1,2,*] and Xiaofeng Peng [1,2,*]

1   Shandong Provincial Engineering and Technical Center of Light Manipulations & Shandong Provincial Key Laboratory of Optics and Photonic Device, School of Physics and Electronics, Shandong Normal University, Jinan 250014, China; 2022020575@stu.sdnu.edu.cn (W.H.); liuleixin530@163.com (L.L.); xianlongliu@sdnu.edu.cn (X.L.)
2   Collaborative Innovation Center of Light Manipulation and Applications, School of Physics and Electronics, Shandong Normal University, Jinan 250358, China
*   Correspondence: yangjian_cai@163.com (Y.C.); xfpeng888@163.com (X.P.)

**Abstract:** In this letter, we conducted a detailed investigation of the statistical properties, such as spectral density, spectral degree of coherence (SDOC), orbital angular momentum (OAM) flux density, and propagation factor $M^2$, of a twisted Gaussian Schell-model (TGSM) beam carrying the cross phase in a turbulent atmosphere. Our findings revealed that atmospheric turbulence induces degeneration of the intensity distribution and spectral degree of coherence of a Gaussian Schell-model beam with the cross phase during propagation, while the twist phase acts as an antidote to degradation. Furthermore, we observed that the z-component of the time-averaged angular momentum flux is determined by the twist phase, whereas the cross phase influences the distribution of the OAM flux density in the beam. Additionally, we explored the variations in the propagation factor $M^2$ of a TGSM beam with the cross phase in a turbulent atmosphere. Notably, we discovered that the deleterious effects of the atmospheric conditions can be mitigated by modulating both the twist and the cross phases. This work contributes valuable insights for information transfer and optical manipulations.

**Keywords:** twist phase; cross phase; turbulent atmosphere; partially coherent beam





## 1. Introduction

Laser beams with variable coherence (i.e., partially coherent beam) have been extensively studied in both theoretical and experimental domains for their significant applications in areas such as inertial confinement fusion, atmospheric optical communication, ghost diffraction, light scattering, sub-Rayleigh imaging, near-field Raman scattering, and atomic cooling [1–8]. Moreover, partially coherent beams with special spatial correlation structures (i.e., correlation structures not satisfying the Gaussian distribution) have further expanded the prospects for the application of partially coherent light beams [9–14] since Gori and Santarsiero introduced a sufficient condition for formulating authentic spatial correlation functions, encompassing both spatially uniform and non-uniform coherence states. For example, it was shown that light beams with spatially varying correlation exhibit extraordinary characteristics, such as locally sharpened and laterally shifted intensity maxima [9]. A light beam also displays a tunable flat profile, a ring shape, and an array distribution in the far field when the correlation structures of the beam satisfy multi-Gaussian functions, Laguerre–Gaussian functions, and the properties of optical coherence lattices, respectively [10–12].

In addition, the twist phase, a specific type of phase different from the usual quadratic phase, can only exist in partially coherent beams and was first highlighted by Simon et al. [15]. Due to its unique characteristics, considerable research has been devoted to studying light beams carrying the twist phase [16–29]. It was found that the beam rotates during propagation, beyond the classical Rayleigh limit, which effectively reduces beam scintillation and

causes its orbital angular momentum to be driven by the twist phase [17,18,25,27]. In addition, studies explored the possibility of introducing the twist phase into a partially coherent beam containing special spatial correlation structures [20,23], and some alternative methods have been proposed to generate partially coherent beams carrying the twist phase [26]. Unlike previous complex methods [27], the most recent methods can not only truly generate twisted partially coherent beams, but also simplify the steps. Based on the aforementioned research, several twisted partially coherent beams with special spatial correlation structures have also been proposed and generated [28,29]. It was show that the twist phase will induce the rotation of the degree of coherence of a twisted partially coherent beam on propagation and influence the spectral degree of polarization distribution. Recently, a novel quadratic phase structure named cross phase has been proposed and investigated. It was found that with the help of the cross phase, one can achieve controllable conversion between Hermite Gaussian and Laguerre Gaussian modes [30], rotating the beam during propagation [31], achieving polygonal shaping and multi-singularity manipulation of optical vortices [32], and measuring the topological charge [33]. Moreover, the cross phase can also be used to enhance the self-healing ability of the beam [34].

On the other hand, as laser beams serve as the primary medium for optical communication, it is very important to study their propagation characteristics in the presence of atmospheric turbulence [2,3,29,35–39]. In comparison to coherent beams, partially coherent beams offer more possibilities to mitigate the adverse effects caused by turbulence [2,3,29,36]. Furthermore, the twist phase and the cross phase can further enhance the turbulence resilience of partially coherent beams [18,37]. To our knowledge, the behavior of partially coherent beams carrying both the twist phase and the cross phase in the presence of atmospheric turbulence has not been studied. In this paper, we introduce a novel partially coherent beam (i.e., a twisted Gaussian Schell-model (TGSM) beam carrying the cross phase) carrying both the twist phase and the cross phase. The evolution of spectral density, SDOC, OAM flux density, and $M^2$-factor of the twisted Gaussian Schell-model (TGSM) beam carrying the cross phase in free space and in a turbulent atmosphere was studied in detail, and some interesting results were found. Our results could be valuable in the fields of information transfer and optical manipulations.

## 2. Propagation Equation of a Twisted Gaussian Schell-Model Beam Carrying the Cross Phase in a Turbulent Atmosphere

It is well known that for scalar partially coherent beams, their statistical properties can be characterized by the cross-spectral density (CSD) function. In this paper, our focus was on the study of the evolution of the properties of a twisted Gaussian Schell-model beam carrying the cross phase in a turbulent atmosphere. Its CSD in the source plane ($z = 0$) is expressed as

$$
\begin{aligned}
W_0(\boldsymbol{r_1}, \boldsymbol{r_2}) &= \exp\left(-\frac{r_1^2 + r_2^2}{4\sigma_0^2}\right) \exp\left[-\frac{(\boldsymbol{r_1} - \boldsymbol{r_2})^2}{2\delta_0^2}\right] \exp[-ik\mu_t x_1 y_2 + ik\mu_t x_2 y_1] \\
&\times \exp[-ik\mu_c x_1 y_1 + ik\mu_c x_2 y_2],
\end{aligned}
\tag{1}
$$

where $\boldsymbol{r}_i = (x_i, y_i)$ represents a transverse position vector in the source plane. $k = 2\pi/\lambda$, $\sigma_0$, and $\delta_0$ represent the wavenumber, the beam width, and the coherence width, respectively. The last two exponential terms represent the twist phase and the cross phase, where $\mu_t$ and $\mu_c$ denote the twist factor and the cross-phase factor used to quantify the magnitude of the twist and the cross phase.

Applying the extended Huygens–Fresnel integral [39], the propagation of the twisted Gaussian Schell-model beam carrying the cross phase can be expressed as

$$
\begin{aligned}
W(\boldsymbol{\rho_1}, \boldsymbol{\rho_2}) &= \frac{k^2}{4\pi^2 z^2} \int_{-\infty}^{\infty} \int_{-\infty}^{\infty} \int_{-\infty}^{\infty} \int_{-\infty}^{\infty} W_0(\boldsymbol{r_1}, \boldsymbol{r_2}) \exp\left[-\frac{ik}{2z}(\boldsymbol{r_1} - \boldsymbol{\rho_1})^2 + \frac{ik}{2z}(\boldsymbol{r_2} - \boldsymbol{\rho_2})^2\right] \\
&\times \langle \exp[\psi(\boldsymbol{r_1}, \boldsymbol{\rho_1}, z) + \psi^*(\boldsymbol{r_2}, \boldsymbol{\rho_2}, z)]\rangle d\boldsymbol{r_1} d\boldsymbol{r_2},
\end{aligned}
\tag{2}
$$

where $\rho_i = (x_i, y_i)$ represents the transverse position vector in the output plane, and $\langle \cdots \rangle$ indicates the ensemble averaging over the turbulent medium and can be formulated as [3,16]

$$\langle \exp[\psi(\boldsymbol{r_1}, \boldsymbol{\rho_1}, z) + \psi^*(\boldsymbol{r_2}, \boldsymbol{\rho_2}, z)] \rangle = \exp\left[ -\frac{1}{\rho_0^2} \left[ (\boldsymbol{\rho_2} - \boldsymbol{\rho_1})^2 + (\boldsymbol{\rho_2} - \boldsymbol{\rho_1}) \cdot (\boldsymbol{r_2} - \boldsymbol{r_1}) + (\boldsymbol{r_2} - \boldsymbol{r_1})^2 \right] \right] \tag{3}$$

where $\rho_0 = \left( 0.545 C_n^2 k^2 z \right)^{-3/5}$, and $C_n^2$ is the structure constant of turbulence. Then, substituting Equations (1) and (3) into Equation (2), the cross-spectral density (CSD) function of the TGSM beam carrying the cross phase in a turbulent atmosphere can be derived as

$$\begin{aligned} W(\boldsymbol{\rho_1}, \boldsymbol{\rho_2}) \quad &= \frac{k^2}{4\pi^2 z^2} \exp\left[ -\frac{1}{\rho_0^2} (\boldsymbol{\rho_2} - \boldsymbol{\rho_1})^2 \right] \exp\left[ -\frac{ik}{2z} \left( \rho_1^2 - \rho_2^2 \right) \right] \sqrt{\frac{\pi}{N_1}} \sqrt{\frac{\pi}{N_2}} \sqrt{\frac{\pi}{N_3}} \sqrt{\frac{\pi}{N_4}} \\ &\times \exp\left[ \frac{1}{4N_1} \left( \frac{ik\rho_{x1}}{z} + \frac{(\rho_{x2} - \rho_{x1})}{\rho_0^2} \right)^2 \right] \exp\left( \frac{M_4^2}{4N_2} \right) \exp\left[ \frac{M_6^2}{4N_3} \right] \exp\left( \frac{M_7^2}{4N_4} \right), \end{aligned} \tag{4}$$

with

$$\begin{aligned} N_1 &= \left( \frac{1}{4\sigma_0^2} + \frac{1}{2\delta_0^2} + \frac{1}{\rho_0^2} + \frac{ik}{2z} \right), N_2 = N_1 + \frac{k^2 \mu_c^2}{4N_1}, \\ N_3 &= \left( N_1^* - \frac{M_1^2}{4N_1} - \frac{M_3^2}{4N_2} \right), N_4 = \left( N_1^* + \frac{k^2 \mu_t^2}{4N_1} - \frac{M_2^2}{4N_2} - \frac{M_5^2}{4N_3} \right), \end{aligned} \tag{5}$$

$$\begin{aligned} M_1 &= \left( \frac{1}{\delta_0^2} + \frac{2}{\rho_0^2} \right), M_2 = \left( M_1 - \frac{k^2 \mu_t \mu_c}{2N_1} \right), M_3 = \left( ik\mu_t - \frac{ik\mu_c M_1}{2N_1} \right), \\ M_4 &= \left( \frac{iky_1}{z} + \frac{y_2 - y_1}{\rho_0^2} + \frac{k^2 \mu_c x_1}{2N_1 z} - \frac{ik\mu_c (x_2 - x_1)}{2N_1 \rho_0^2} \right), \end{aligned} \tag{6}$$

$$\begin{aligned} M_5 &= \left( ik\mu_c - \frac{ik\mu_t M_1}{2N_1} + \frac{M_2 M_3}{2N_2} \right), \\ M_6 &= \left\{ \frac{ikx_1 M_1}{2N_1 z} + \frac{M_1 (x_2 - x_1)}{2N_1 \rho_0^2} - \frac{ikx_2}{z} - \frac{(x_2 - x_1)}{\rho_0^2} + \frac{M_3 M_4}{2N_2} \right\}, \\ M_7 &= \left( \frac{M_5 M_6}{2N_3} + \frac{k^2 \mu_t x_1}{2N_1 z} - \frac{ik\mu_t (x_2 - x_1)}{2N_1 \rho_0^2} - \frac{iky_2}{z} - \frac{(y_2 - y_1)}{\rho_0^2} + \frac{M_2 M_4}{2N_2} \right). \end{aligned} \tag{7}$$

In the above derivations, we used the following expansion and integral formulae

$$\int_{-\infty}^{\infty} \exp\left( -p^2 x^2 \pm qx \right) dx = \exp\left( \frac{q^2}{4p^2} \right) \frac{\sqrt{\pi}}{p}, \quad \left( \mathrm{Re}\, p^2 \geq 0 \right) \tag{8}$$

The spectral density and the spectral degree of coherence (SDOC) of the TGSM beam carrying the cross phase in the output plane were then obtained as

$$S(\boldsymbol{\rho}, z) = W(\boldsymbol{\rho}, \boldsymbol{\rho}). \tag{9}$$

$$\gamma(\boldsymbol{\rho_1}, \boldsymbol{\rho_2}) = \frac{W(\boldsymbol{\rho_1}, \boldsymbol{\rho_2})}{\sqrt{W(\boldsymbol{\rho_1}, \boldsymbol{\rho_1}) W(\boldsymbol{\rho_2}, \boldsymbol{\rho_2})}}. \tag{10}$$

Utilizing Equations (4)–(10), the characteristics of the evolution of both the spectral intensity and the SDOC in a turbulent atmosphere can be efficiently investigated.

## 3. Theoretical Analysis of the Statistical Properties of a TGSM Beam Carrying the Cross Phase in a Turbulent Atmosphere

On the basis of the obtained formulas in the last section, we studied the propagation properties of the TGSM beam carrying the cross phase in turbulent conditions. The related parameters were $\lambda$ = 1550 nm, $\sigma_0$ = 1 cm, and $\delta_0$ = 1 cm, unless otherwise specified.

We studied the evolution of the spectral density of a TGSM beam carrying the cross phase at different distances in free space, as shown in Figure 1. We observed that with the increase in the distance, the beam profile firstly transformed from a Gaussian distribution to an elliptical distribution due to cross-phase effects and then, as the distance

further increased, reverted to a Gaussian distribution in the far field. When the beam simultaneously possessed twist and cross phases, the twist phase not only caused the beam to rotate during the transmission process, but also maintained its elliptical distribution. Additionally, the rotation direction of the beam was determined by the chirality of the twist phase. In order to further investigate the relationship between beam eccentricity and these phases, we considered the eccentricity of a TGSM beam carrying the cross phase at different distances in free space, with different values of $\mu_t$ and $\mu_c$, as shown in Figure 2. The eccentricity is denoted as $e = \sqrt{1 - (b/a)^2}$, with a and b being the semi-major and the semi-minor axis of the ellipse, respectively. The shape is perfectly circular when e = 0 and becomes elongated or elliptical as e increases. In Figure 2, one can see that with the increase in the distance, the beam gradually transitions from a Gaussian distribution on the source plane (i.e., e = 0) to an elliptical distribution. The eccentricity of the beam reaches a maximum value (which increases with the augmentation in the cross phase); with a further increase in the transmission distance, the beam changes from an elliptical distribution back to a Gaussian distribution. This phenomenon can be elucidated by examining the orbital angular momentum distribution of the beam. Furthermore, when a twist phase was introduced, the beam effectively maintained its elliptical distribution, as illustrated in Figure 1(a2–d3).

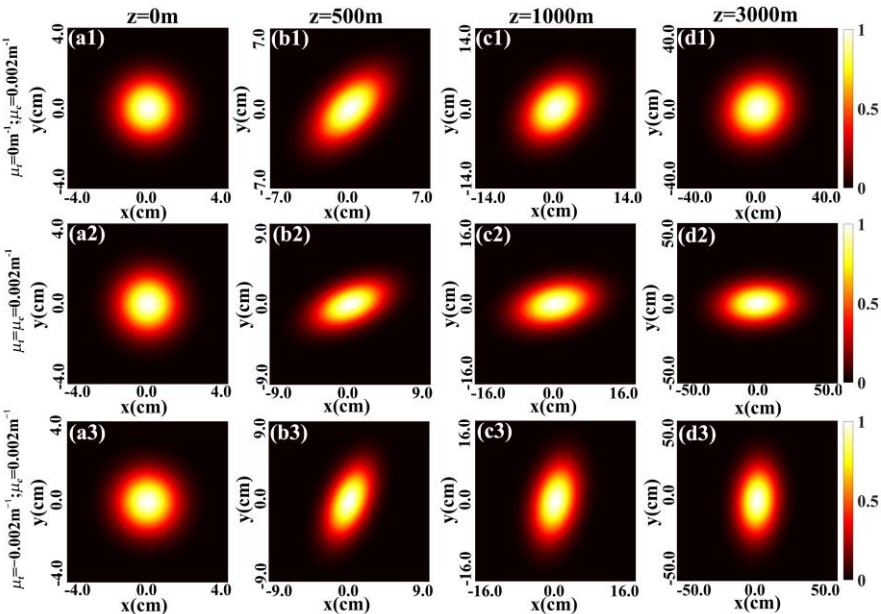

**Figure 1.** Normalized spectral density of a TGSM beam (**a1–d1**) without the cross phase, (**a2–d3**) carrying the cross phase, at different distances in free space ($C_n{}^2 = 0$), with different values of $\mu_t$ and $\mu_c$.

Figure 3 shows the normalized spectral density of a TGSM beam carrying the cross phase at different distances in a turbulent atmosphere ($C_n{}^2 = 5 \times 10^{-14}$ m$^{-2/3}$), with different values of $\mu_t$, $\mu_c$. One can see that a turbulent atmosphere accelerates the process of the beam transitioning from an elliptical distribution to a Gaussian distribution, i.e., a turbulent atmosphere leads to the degeneration of the spectral density distribution. In contrast, the twist phase plays a role in impeding this transformation, which means that one can use the twist phase to control the beam profile distribution on propagation.

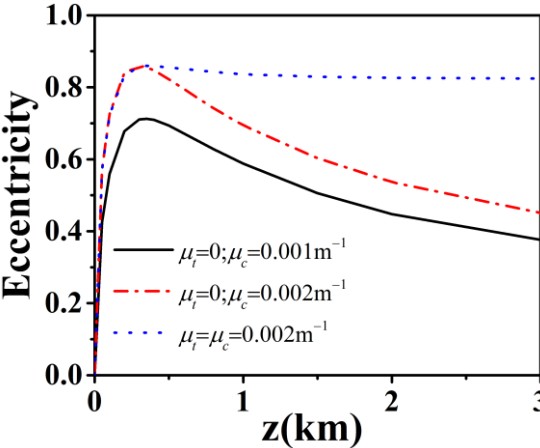

**Figure 2.** Eccentricity of a TGSM beam carrying the cross phase at different distances in free space ($C_n{}^2 = 0$), with different value of $\mu_t$ and $\mu_c$.

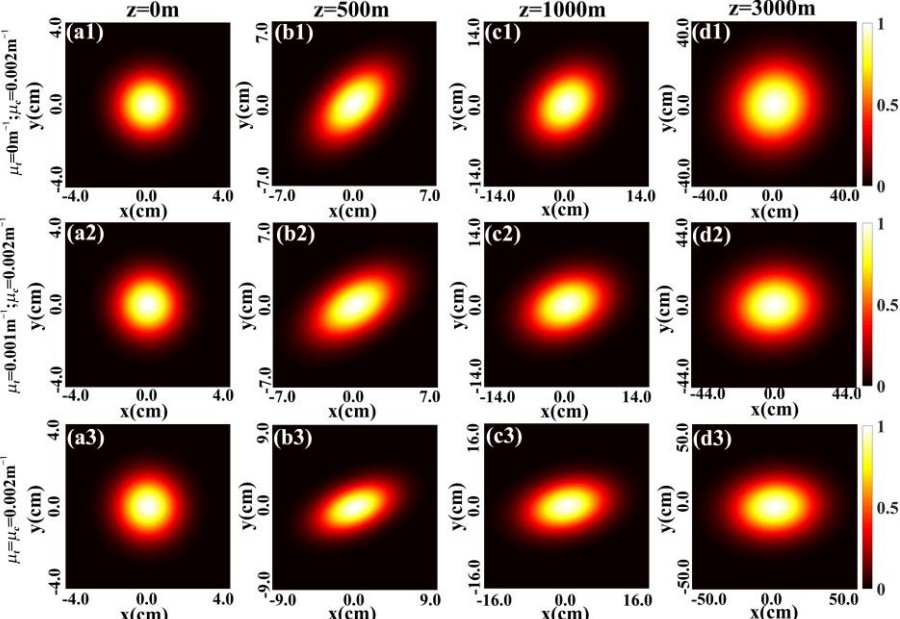

**Figure 3.** Normalized spectral density of a TGSM beam carrying the cross phase at different distances in a turbulent atmosphere ($C_n{}^2 = 5 \times 10^{-14}$ m$^{-2/3}$), with different values of $\mu_t$ and $\mu_c$.

As shown in Figure 4, we studied the evolution of the SDOC of a TGSM beam carrying the cross phase at different distances in free space. Similarly to the evolution of the spectral density, when there was no twist phase, the SDOC of the beam transitioned from a Gaussian distribution to an elliptical distribution, eventually returning to a Gaussian distribution. In contrast, in the presence of the twist phase, the SDOC of the beam could effectively maintain its elliptical distribution. Furthermore, the twist phase also induced the rotation of the SDOC on propagation, while the rotation direction was precisely opposite to the direction of the beam spectral density. Figure 5 demonstrates the normalized SDOC of the beam at z = 1000 m with different values of $C_n{}^2$, $\mu_t$, and $\mu_c$. It was found that with the increase in the turbulence strength $C_n{}^2$, the distribution of the SDOC gradually transformed from an elliptical distribution to a Gaussian distribution, while the twist phase hindered this transformation. In order to further explore the influence of twist phase and cross phase on the evolution characteristics of the beam, we investigated the distribution of the beam's

orbital angular momentum. The OAM flux density of a partially coherent beam can be expressed as [40]

$$L_d(\boldsymbol{\rho}) = -\frac{\varepsilon_0}{k} I_m \left[ \left( y_1 \partial_{x_2} - x_1 \partial_{y_2} \right) W(\boldsymbol{\rho}_1, \boldsymbol{\rho}_2) \right]_{\rho_1 = \rho_2 = \rho'} \tag{11}$$

where $\varepsilon_0$ is the permittivity in vacuum. $I_m$ stands for the imaginary part, and $\partial_{\alpha_2}$ denotes the partial derivative with respect to $\alpha_2$. Substituting Equation (4) into Equation (10), the OAM flux density of the TGSM beam carrying the cross phase could be analyzed.

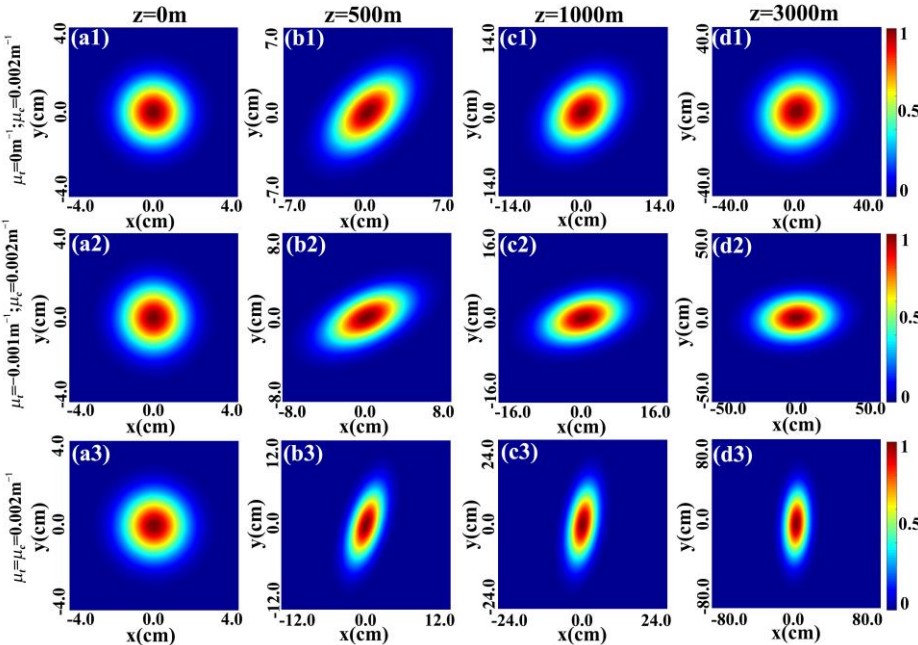

**Figure 4.** Normalized SDOC [$|\gamma(\boldsymbol{\rho}_1, 0)|$] of a TGSM beam (**a1–d1**) without the cross phase, (**a2–d2**) carrying the cross phase at different distances in free space ($C_n^2 = 0$), with different values of $\mu_t$, and $\mu_c$.

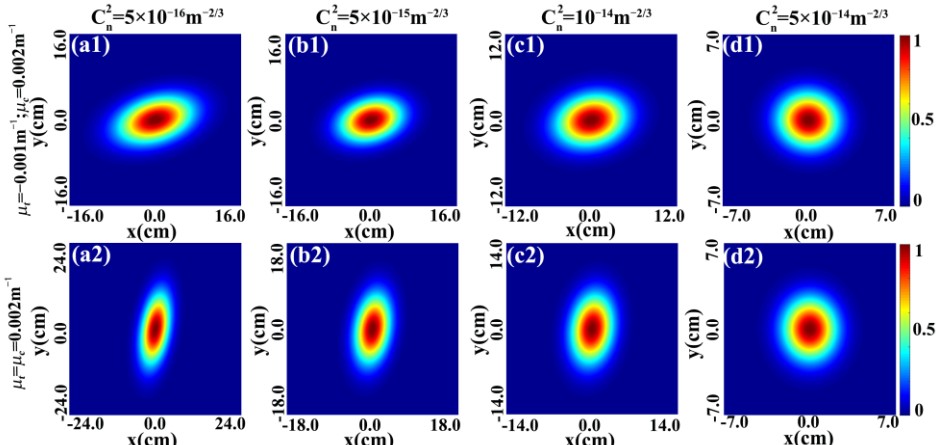

**Figure 5.** Normalized SDOC [$|\gamma(\boldsymbol{\rho}_1, 0)|$] of a TGSM beam carrying the cross phase to z = 1000 m in a turbulent atmosphere with different values of $C_n^2$, $\mu_t$, and $\mu_c$.

Figure 6 shows the distribution of the OAM flux density of the TGSM beam carrying the cross phase in free space. It was found that when the beam did not carry these phases, the OAM flux density of the beam was zero, as reported in [41]. In contrast, when the beam carried the twist phase, the OAM flux density of the beam exhibited a circular distribution. Moreover, it was interesting to observe that, although the time-averaged

angular momentum flux of the beam was determined by the twist phase, the cross phase influenced the distribution of the beam's orbital angular momentum. Furthermore, during the propagation process, there was a reversal in the positive and negative regions, which also explains why the spectral density of the beam in Figure 1 changes from elliptical to Gaussian.

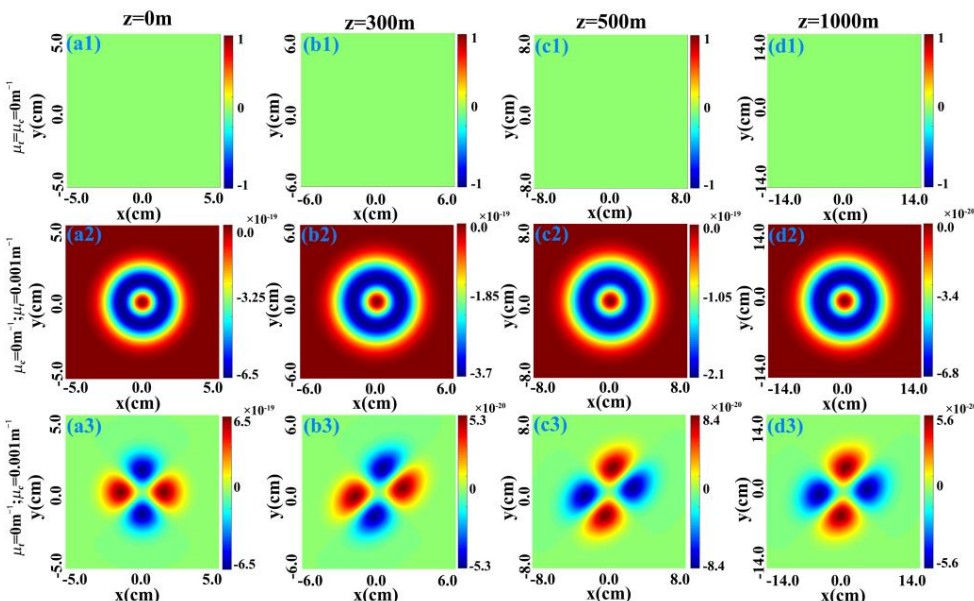

**Figure 6.** Variation of the OAM flux density of the TGSM beam carrying the cross phase in free space ($C_n{}^2 = 0$), with different values of $\mu_t$ and μc.

As shown in Figure 7, we investigated the variation of the OAM flux density of the beam in a turbulent atmosphere. We found that when the beam carried only the cross phase, the evolution of the OAM flux density was similar to that in free space. When the beam simultaneously carried both phases with equal magnitudes, the orbital angular momentum of the beam transformed into two spots, whose orientation was determined by the chirality of the twist and cross phases. When the chirality of the two phases was the same, the initial orientation was along the y-axis. Conversely, when the phases had opposite chirality, the orientation was along the x-axis. Moreover, with the increase in the propagation distance, the two spots initially converged and then separated again into two distinct spots, and the orientation was perpendicular to the initial direction. This phenomenon has potential applications in cases where information is encoded in the orbital angular momentum.

To further investigate the roles of these two phases in turbulence resistance, we studied the evolution of the propagation factor M$^2$ in the presence of atmospheric turbulence, according to [29]

$$M^2 = k\sqrt{\langle r^2 \rangle \langle \theta^2 \rangle - \langle \boldsymbol{r} \cdot \boldsymbol{\theta} \rangle^2}, \tag{12}$$

where $\langle r^2 \rangle, \langle \boldsymbol{r} \cdot \boldsymbol{\theta} \rangle$, and $\langle \theta^2 \rangle$ denote the second-order moments of the beam. With the help of the propagation law of the second-order momentum of a partially coherent beam in turbulent atmosphere [42], the second-order moments of the beam can be expressed as

$$\begin{aligned}
\langle r^2 \rangle &= 2\sigma_0^2 + \left( \frac{1}{2k^2\sigma_0^2} + \frac{2}{k^2\delta_0^2} + 2\mu_t^2\sigma_0^2 + 2\mu_c^2\sigma_0^2 \right)z^2 + \frac{4}{3}\pi^2 T z^3, \\
\langle \theta^2 \rangle &= \frac{1}{2k^2\sigma_0^2} + \frac{2}{k^2\delta_0^2} + 2\mu_t^2\sigma_0^2 + 2\mu_c^2\sigma_0^2 + 4\pi^2 T z, \\
\langle \boldsymbol{r} \cdot \boldsymbol{\theta} \rangle &= \left( \frac{1}{2k^2\sigma_0^2} + \frac{2}{k^2\delta_0^2} + 2\mu_t^2\sigma_0^2 + 2\mu_c^2\sigma_0^2 \right)z + 2\pi^2 T z^2,
\end{aligned} \tag{13}$$

where $T = 3/\pi^2 k^2 z \rho_0^2$. Substituting Equation (12) into Equation (10), the propagation factor M$^2$ can be analyzed.

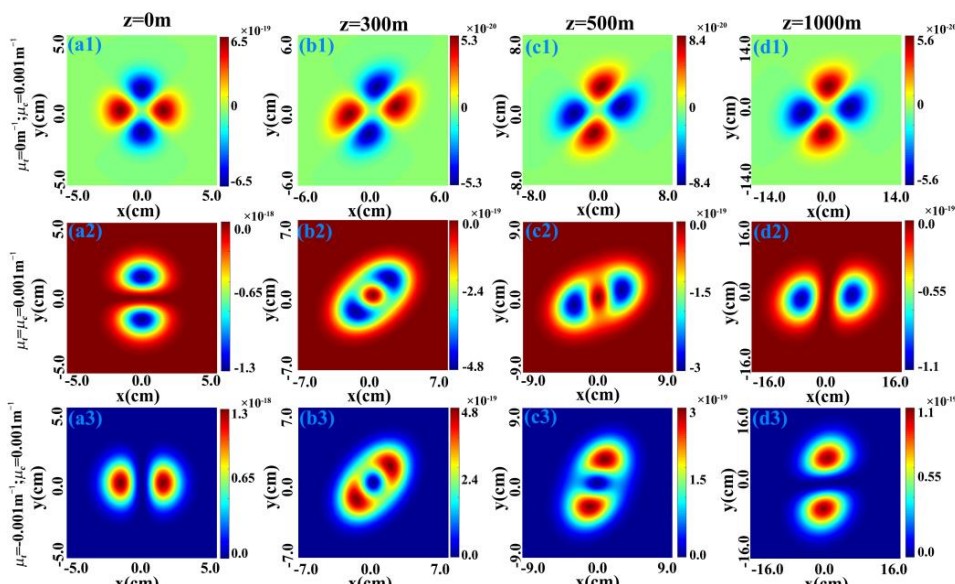

**Figure 7.** Variation of the OAM flux density of the TGSM beam carrying the cross phase in a turbulent atmosphere ($C_n{}^2 = 5 \times 10^{-14}$ m$^{-2/3}$), with different values of $\mu_t$ and $\mu_c$.

In Figure 8, it can be seen that turbulence induced an increase in the propagation factor $M^2$ of the beam during the propagation. We found that with the increase in the twist factor $\mu_t$ or in the cross-phase factor $\mu_c$, the propagation factor decreased. And under the combined action of the twist phase and the cross phase, the $M^2$ factor, previously increased by turbulence, could further decrease. The dependence of the $M^2$ factor on coherence width is shown in Figure 8d, where the distance was fixed at z = 3000 m. One can see that the $M^2$ factor decreased rapidly as the coherence width decreased. This phenomenon can be explained by the fact that the coherence width plays a dominant role in determining the value of the twist factor and has an effect on the $M^2$ factor.

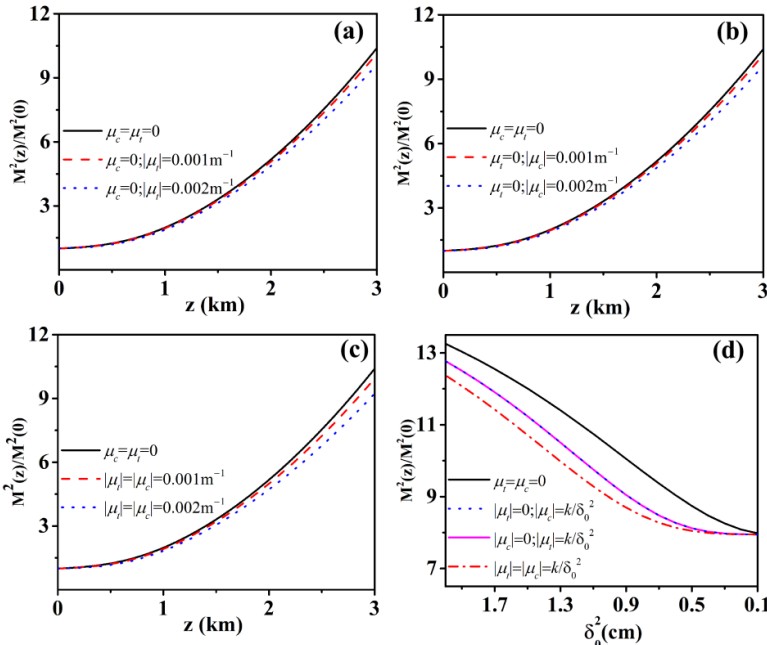

**Figure 8.** Variation of the normalized propagation factor $M^2$ of the TGSM beam carrying the cross phase in turbulent atmosphere ($C_n{}^2 = 5 \times 10^{-14}$ m$^{-2/3}$), with different value of (**a**) the twist factor and the cross-phase factor (**a**) the twist factor $\mu_t$, (**b**) the cross-phase factor $\mu_c$, (**c**) the combine action of the twist factor and the cross-phase factor and (**d**) the coherence width $\delta_0$.

## 4. Discussion

The twist phase and the cross phase, as crucial modulation parameters for partially coherent beams, impart novel physical characteristics to beams. However, no prior research is available on the combined action of these two phases. In this study, we investigated the transmission statistical properties of beams under the combined influence of these two phases. It is important to note that, despite the beam carrying both types of phases, the constraints on the twist phase were still determined by its coherence structure [23]. For Gaussian Schell-model correlated beams, the twist factor should satisfy the condition $\mu_t \leq 1/k\delta_0^2$. In addition, it is crucial to emphasize that while the analysis presented in this study is entirely theoretical, the practical generation of this beam is feasible. To achieve this, we can firstly generate TGSM beams by using a random complex screen [26]. Subsequently, the cross phase can be imparted to the beam using a spatial light modulator (SLM). Furthermore, as demonstrated in references [18,42], the modulation of the twist phase or of the cross phase in a partially coherent beam was proven to mitigate turbulence-induced scintillation. Consequently, we anticipate that a twisted Gaussian Schell-model beam incorporating the cross phase would offer advantages compared to a scalar TGSM beam or to a GSM beam with the cross phase, further enhancing the reduction of turbulence-induced scintillation.

## 5. Conclusions

In summary, we thoroughly investigated the statistical properties of the TGSM beam carrying the cross phase in both free space and turbulent atmosphere, utilizing the derived analytical formula. Our results illustrate that, in addition to cause the beam to rotate, the twist phase can maintain the elliptical distribution of the beam profile on propagation. Moreover, the twist phase can effectively mitigate the degradation of the beam spectral density and SDOC caused by turbulence. Furthermore, it is interesting to note that the distribution of the OAM flux density of the beam can be controlled by modulating these phases, which has potential applications in information transfer. In addition, using these two phases can effectively improve the quality of the propagation factor $M^2$. Our findings are expected to have applications in information transfer and optical manipulations.

**Author Contributions:** Conceptualization, L.L. and X.P.; methodology, L.L., Y.C. and X.P.; validation, W.H. and X.P.; formal analysis, L.L., Y.C. and X.P.; investigation, W.H., L.L., X.L., Y.C. and X.P.; resources, L.L., Y.C. and X.P.; data curation, W.H.; writing—original draft preparation, W.H. and X.P.; writing—review and editing, L.L. and Y.C. and X.P.; visualization, W.H.; supervision, L.L., Y.C. and X.P.; project administration, L.L. and X.P.; funding acquisition, Y.C. All authors have read and agreed to the published version of the manuscript.

**Funding:** This research was funded by the National Key Research and Development Program of China (2019YFA0705000, 2022YFA1404800); the National Natural Science Foundation of China (12104263, 12192254, 11974218, 92250304, 12304365, 12274268); the Local Science and Technology Development Project of the Central Government (YDZX20203700001766); the Natural Science Foundation of Shandong Province (ZR2021QA093, ZR2023QA088); and the China Postdoctoral Science Foundation (2022M721993).

**Institutional Review Board Statement:** Not applicable.

**Informed Consent Statement:** Not applicable.

**Data Availability Statement:** The data underlying the results presented in this paper are not publicly available at this time but may be obtained from the authors upon reasonable request.

**Conflicts of Interest:** The authors declare no conflicts of interest.

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
