# Peer review of "Statistical Properties of a Twisted Gaussian Schell-Model Beam Carrying the Cross Phase in a Turbulent Atmosphere"

_photonics, doi:10.3390/photonics11020124_

Round 1

Reviewer 1 Report

Comments and Suggestions for Authors

In this manuscript (MS), the statistic properties (including spectral density, SDOC, OAM flux density and propagation factor) of a twisted Gaussian Schell-model beam carrying cross phase are studied in a turbulent atmosphere. Several interesting phenomena are observed and analyzed, such as that the total OAM flux density is determined only by the twist phase, but the local density can be influenced by the cross phase. The finding in this MS could be very valuable in controlling/designing the partial coherent beams in imaging and communication.  While I have some concerns listed below, and with these concerns addressed I believe the MS can be considered acceptable for publication.

1. In Figs. 2 and 8, the numbers along axes seem too big, while the words in the frames are too small.

2. In the caption of Fig. 3, the Cn and \mu_t, \mu_c are not displayed in correct form.

3. It says “When the chirality of the two phases is the same, the initial orientation is along the y-axis. Conversely, when they have opposite chirality, the orientation is along the x-axis.” While in third line of Fig. 7, only the case of \mu_t<0 and \mu_c>0 is shown, so how about the case of \mu_t>0 and \mu_c<0? Whether is the rule found in this MS still valid?

Reviewer 2 Report

Comments and Suggestions for Authors

In this manuscript, a twisted Gaussian Schell-model beam carrying cross phase in a turbulent atmosphere are discussed. The spectral density, the spectral degree of coherence, the orbital angular momentum flux density and the propagation factor M2 are investigated in detail. In my opinion, the paper is clearly written and well organized. But some issues need the authors to add and address, which may helpful for promoting this work to broadly audience.

1) The conclusion ...the total OAM flux density of the beam is determined by the twist phase, the cross phase influences the distribution... from Figure 6 is written in the line 169. The case when the twist phase factor and cross phase factor are 0 in the first line of Figure 6 is special. The distribution of OAM flux density are totally different when the twist phase factor or the cross phase factor is not zero. Hence, it is not easy to find the determinate parameter,

2) It is said that We find that when the beam carries only cross phase, the evolution of the OAM flux density is similar to that in free space. in the line 179. What about the beam carries only twist phase comparing with the case in the second line of Fig 6?

3) The caption of Fig 8 with different value of mt, mc seems to be ,   respectively.

...the twist factor mt... also needs to be corrected.

Reviewer 3 Report

Comments and Suggestions for Authors

In this work, the authors introduced a new general model of a partially coherent beam, a kind of twisted Gaussian Schell-model beam (TGSM) carrying a cross phase, and studied the evolution of its statistical properties, such as the spectral density (SD), the spectral degree of coherence (SDOC), the orbital angular momentum (OAM) flux density, as well as the beam propagation factor (M2-factor) in free space and under the turbulent atmosphere. The authors used the Huygens-Fresnel integral diffraction to derive the analytical expression of the cross-spectral density and obtained the corresponding SD and the SDOC of the TGSM with a cross phase. Then, based on the obtained formulas, they analyzed numerically the statistical properties of the beam in free space and a turbulent atmosphere. They found that the twist and cross phase modulations can maintain the elliptical distribution of the beam profile upon propagation and reduce the deterioration of the beam caused by atmospheric turbulence. In addition, they show that the-phase modulation can allow for control of the OAM, and improve the propagation quality (i.e., reducing M2-factor) of the beam. The idea of the work, considering a beam with both twist and cross-phase modulation, is intuitive and of interest. The analytical derivations seem correct, and the numerical calculations are clearly illustrated. The paper is well written. The obtained results could be beneficial to practical applications in optical communication and micromanipulation of particles. So, I think, this paper could be published in Photonics in its current form. However, I suggest the following comment points should be taken into account in the revised manuscript.

1-In the introduction part, line 67, the authors should add “the M2-factor ”:

…., the SDOC and the OAM flux density of a   should be: the SDOC the OAM flux density, M2-factor of a…

2-Line 78: ri = (xi0, yi0) should be ri = (xi, yi).

3-Line 85: ri = (xi, yi) should be ri = (xi, yi).

4- Line 130: Figure 3 (in the legend of Fig. 3) should be in bold.

5- Line 210: “4. Conclusions” should be “4. Conclusion”.

Reviewer 4 Report

Comments and Suggestions for Authors

Wenshuo Hou, Leixin Liu, Xianlong Liu, Yangjian Cai, and Xiaofeng Peng Statistical properties of a twisted Gaussian Schell-model beam carrying cross phase in a turbulent atmosphere

 This paper presents a numerical analysis of the statistical properties of the TGSM beam carrying both twist phase and cross phase in free space and turbulent atmosphere using an analytical formula for scalar CSD. The evolution of the spectral density, spectral degree of coherence, and OAM flux density of a twisted Gaussian Shell-model (TGSM) beam carrying cross phase in free space and turbulent atmosphere is studied in the paraxial approximation. It is shown that the twist and cross phases not only cause the beam to rotate during propagation, but also maintain its elliptical distribution. The results of this work may be of interest to specialists.

 Specific comments 

11.     Expression (4) is obtained in the paraxial approximation. It would be useful for readers to discuss the effect of nonparaxiality during propagation.

22.   Line 85: a typo in the designation of the transverse position vector in the output plane.

33. The results are obtained for an incident scalar wave. Light sources are usually polarized. How will the results change if polarization is taken into account?

44. The authors claimed a detailed study of the spectral degree of coherence (SDOC). However, the results are not presented. Some numerical or graphical illustration of the SDOC behavior with distance would be useful.

5 5. Will the results be observed for beams with other correlation structures, for example, for Bessel-correlated beams?

The manuscript can be accepted for publication in Photonics after making revisions mentioned above. 
